# Work Placement and Job Satisfaction in Long-Term Childhood Cancer Survivors: The Impact of Late Effects

**DOI:** 10.3390/cancers14163984

**Published:** 2022-08-18

**Authors:** Margherita Dionisi-Vici, Alessandro Godono, Anna Castiglione, Filippo Gatti, Nicoletta Fortunati, Marco Clari, Alessio Conti, Giulia Zucchetti, Eleonora Biasin, Antonella Varetto, Enrico Pira, Franca Fagioli, Enrico Brignardello, Francesco Felicetti

**Affiliations:** 1Transition Unit for Childhood Cancer Survivors, Città della Salute e della Scienza Hospital, 10126 Turin, Italy; 2Clinical Psychology Unit, Città della Salute e della Scienza Hospital, 10126 Turin, Italy; 3Department of Public Health and Pediatric Sciences, University of Turin, 10126 Turin, Italy; 4Unit of Clinical Epidemiology, Città della Salute e della Scienza Hospital, 10126 Turin, Italy; 5Division of Paediatric Onco-Haematology, Stem Cell Transplantation and Cellular Therapy, Città della Salute e della Scienza Hospital, 10126 Turin, Italy

**Keywords:** childhood cancer survivors, work placement, occupation, job satisfaction, late effects, satisfaction profile

## Abstract

**Simple Summary:**

Due to previous cancer and its treatments, long-term childhood cancer survivors (CCS) are at risk of developing medical comorbidities, as well as socioeconomic vulnerability. Above all, work placement, financial independence, and job satisfaction can represent burdensome areas for some CCS. Regarding this topic, it is important to consider the variety of country-specific educational and vocational systems. The aim of this study is to provide a description of the occupational status of CCS in the Italian socio-economic scenery and to evaluate the association between late effects and unemployment and late effects and job satisfaction. Our data contribute to the description of the different country scenery of work placement for CCS in Europe and agree with previous literature on the impact of severe late effects on occupation. Future research in this field can be focused on interventions to improve CCS in obtaining jobs suitable for their health.

**Abstract:**

Late effects of cancer and its treatments during childhood or adolescence can impact work placement and increase the risk of unemployment. The aim of this study is to describe the work placement and the perceived job and economic satisfaction of long-term childhood cancer survivors (CCS). Jobs have been categorized according to the International Standard Classification of Occupations version 08 (ISCO-08), and satisfaction has been evaluated through the Satisfaction Profile (SAT-P). Out of 240 CCS (female = 98) included: 53 were students, 46 were unemployed and 141 were employed. Within unemployed survivors, 89.13% were affected by late effects (*n* = 41). The presence of at least one severe late effect was significantly associated with the probability of unemployment (OR 3.21; 95% CI 1.13–9.12, *p* < 0.050), and having any late effect was inversely related to the level of satisfaction of the financial situation of unemployed CCS (b −35.47; 95% CI −59.19, −11.74, *p* = 0.004). Our results showed that being a survivor with severe comorbidities has a significantly negative impact on occupation and worsens the perception of satisfaction of economic situations. Routinary follow-up care of CCS should include the surveillance of socioeconomic development and provide interventions, helping them to reach jobs suitable for their health.

## 1. Introduction

Due to the success of treatment protocols for pediatric tumors over the past decades, the 5-year survival rate is now exceeding 80%. As a result, the population of long-term childhood cancer survivors (CCS) is constantly growing. Monitoring and understanding the long-term consequences that CCS can develop through their life is relevant for public health [1]. Besides the possible onset of physical late effects induced by cancer treatments, CCS are at risk of developing psychological and socioeconomic vulnerability [2,3,4,5].

Key developmental tasks such as educational achievements, employment, financial independence, and job satisfaction are crucial factors for a good quality of life (QoL), for CCS as well as for their healthy peers [4,6,7]. However, due to their previous cancer, CCS may have greater difficulty coping with challenges that entering adulthood implies. This can lead to a prolonged dependency on parents, lengthy interruption of education or to a delayed professional employment, leading to financial problems [3].

Although many CCS have an overall good QoL, others refer a lower satisfaction particularly among socioeconomic areas [3,4,6,7]. As a consequence, monitoring the subjective perception of satisfaction among educational and occupational progress should be included in their routine follow-up. 

The literature on CCS’ employment is heterogeneous, not only due to the variability in country-specific educational systems, but also because of the differences in populations included in the studies (e.g., primary cancer diagnoses, sample sizes, comparison groups etc.) [8]. Although the occupational rate for many long-term CCS seems comparable to those of the general population [1,8], some factors can represent a risk for being unemployed. Furthermore, the employability and/or highly skilled occupation in CCS may be impacted by their previous cancer history and treatments (e.g., younger age at diagnosis, brain tumor malignancies, cranial radiation). Being affected by specific late effect (such as neurocognitive impairment, musculoskeletal disabilities, or psychiatric diagnosis) or by a high number of medical co-morbidities can also impact the occupational rate of CCS [1,2,9,10,11]. Conversely, no significant differences in the risk of unemployment for reasons unrelated to health between survivors, siblings, and population comparisons exist [1]. 

Besides these difficulties, young adult survivors can also experience a positive attitude toward life after overcoming cancer that can lead to faster growing and a feeling of higher maturity compared with their peers [7].

Recent studies in this field focus on the British or Nord Europe CCS population, but a description of the Italian situation is missing [1,12,13].

The aim of this study is to describe the work placement of CCS followed at the Transition Unit for Childhood Cancer Survivors based in Turin, Italy, and to evaluate associations between late effects and occupational rate as well as late effects and job satisfaction.

## 2. Materials and Methods

We include in this study all subjects with a follow-up visit at the Transition Unit, a specialized, adult-focused, out-patient clinics for CCS, [14] between September 2018 and September 2019 with age > 18 years and <35 years, a previous cancer diagnosis at age < 18 years and off-therapy for at least 5 years.

Since 2006, within the Città della Salute e della Scienza Hospital, a multidisciplinary Hospital in Turin (Piedmont, Italy), operates the Transition Unit for childhood cancer survivors. According with our protocols, when they are aged over 18 years and off-therapy for at least 5 years, CCS previously cured for cancer at the Pediatric Oncology Unit of the Hospital are transitioned to this unit to continue their long-term follow-up [14]. For patients with cognitive impairment, severe psychiatric disorders, or conditions otherwise hampering the filling in of the questionnaire (e.g., blindness or lack of Italian language understanding), we only collected clinical and occupational data.

Late effects have been grouped using the St Jude Lifetime Cohort Study (SJLIFE) modified version of the National Cancer Institute’s Common Terminology Criteria for Adverse Events (CTCAE) version 4.03 [15].

Jobs have been categorized according to the International Standard Classification of Occupations version 08 (ISCO-08) [16] into 10 major groups: (1) managers, (2) professionals, (3) technicians and associate professionals, (4) clerical support workers, (5) services and sales workers, (6) skilled agricultural/forestry and fishery workers, (7) craft and related trades workers, (8) plant and machine operators and assemblers, (9) elementary occupations, (10) armed forces occupations. Each major group is further organized into sub-major and minor units. Criteria used for the classification were the level of skill and specialization required to perform the tasks of the specific occupation.

Job satisfaction has been evaluated through the Satisfaction Profile (SAT-P) [17,18,19], a questionnaire that investigates the subjective satisfaction in several domains of daily life. For each item, the subject must evaluate his personal satisfaction in the last month on a 10 cm horizontal scale, ranging from “extremely dissatisfied” to “extremely satisfied”. Higher scores indicate better satisfaction (range 0–100). For this study, only the “Work” scale has been considered (it includes the following items: *type of work, organization of work, professional role, work productivity and financial situation*). Non-working participants at the time of the study needed to cross out only the financial situation item. Students needed to answer the questionnaire considering their study activity as their main job.

The present study complies with the Declaration of Helsinki and was approved by the competent Ethical Committee (protocol number 0098534). A written informed consent was obtained from all participants.

### Statistical Methods

Socio-demographic and clinical characteristics were summarized using absolute and relative frequencies. Age at study was categorized into 3 classes (18–24, 25–29, 30–35 yrs) and age at the first cancer diagnosis in 3 classes (0–4; 5–9; 10–18 yrs). In order to assess the association between unemployment and late effect, we performed multivariate logistic models where we included as covariate sex, age at study time and presence of late effects. We considered late effects both as dichotomous variables (presence vs. absence of late effects) and as ordinal variables (no late effects, at least one moderate and no late effect, at least one severe late effect).

In order to assess association between job satisfaction and the presence of late effects, we performed multivariate linear regression models, including as covariate sex, age, occupational status, late effects and a term interaction of the two last variables. In this way, we estimated the association between late effect and job satisfaction separately in employed and unemployed patients, and we could test if the association was different according to the occupational status.

As sensitivity analysis, we reperformed all models in employed and unemployed CCS, excluding students.

## 3. Results

### 3.1. Socio-Demographic and Clinical Characteristics of the Sample (n = 240)

During a regular follow-up visit at the Transition Unit for Childhood Cancer Survivors, 240 CCS accepted to participate in the study. The inclusion process of participants is reported in Figure 1.

At the time of the study, 39.17% of participants was aged between 18 and 24 years. One hundred and fifteen CCS (47.92%) were aged between 10 and 18 years at the time of cancer diagnosis. Hematologic malignancies were the most frequent diagnoses (72.5%; *n* = 174), followed by brain tumors (12.08%; *n* = 29) and sarcomas (10.83%; *n* = 26). Fifty-two CCS (21.67%) did not have any late effect, whereas moderate and severe late effects were recorded in 37.50% (*n* = 90) and 40.83% (*n* = 98) of enrolled CCS, respectively. 

Fifty-three CCS were students, 46 unemployed and 141 employed. At least a late effect was found in 89.13% of unemployed survivors, in 75.47% of student and in 75.89% of employed CCS (Table 1).

The distribution of occupation categories according to the ISCO-08 classification is shown in Table 2. Among employed CCS, services and sales workers were the most frequent occupations (27.66%; *n* = 39), followed by technicians and associate professionals (24.82%; *n* = 35). Clerical support workers were the less represented (0.71%; *n* = 1).

### 3.2. Unemployment

The presence of any late effect was associated with the probability of unemployment (OR 2.61; 95% CI 0.96–7.08; *p* = 0.060), and the result was confirmed after exclusion of students (OR = 2.96; 95% CI 1.06–8.26; *p* < 0.050) (Table 3). Moreover, the presence of at least one severe late effect was significantly associated with the probability of unemployment both considering the total sample (*n* = 240; OR 3.21; 95% CI 1.13–9.12, *p* < 0.050), as well as comparing only employed and unemployed participants (*n*= 187; OR 3.69; 95% CI 1.25–10.82, *p* < 0.050) (Appendix A).

### 3.3. Satisfaction of Job and Financial Situation (n = 205)

Mean level of satisfaction of job and financial situation did not change according to the presence of late effects in employed or students CCS (without late effects = 68.16 [62.48–73.83] vs. with late effects = 70.36 [67.39–73.34]). In unemployed CCS, the satisfaction of job was equal to financial situation. In these subjects, satisfaction was lower in the presence of late effects (without late effects = 74.80 [45.74–103.86] vs. with late effects =39.68 [27.51–51.85]) (Figure 2).

Multivariate analysis confirmed these results: particularly, the presence of late effects had a negative impact on job satisfaction for unemployed CCS (b −35.94; 95% CI −55.51, −16.37, *p* < 0.001), but no effect in student or employed CCS (b 1.45, 95%CI −5.49, 8.39). Similarly, being affected by any late effect was inversely related to the level of satisfaction of financial situation of unemployed CCS (b −35.47; 95% CI −59.19, −11.74, *p* = 0.004). Analogous results were obtained excluding students from analyses (Table 4 and Table 5).

## 4. Discussion

The data suggest that in CCS, there is an association between the presence of late effects, particularly when severe, and unemployment.

Employment rates observed in our study did not markedly differ from those found in the healthy general population [20]. We found a rate of employment of 58.75% (and of 19.17% for unemployment) and an occupation rate of 75.41% after exclusion of students. In young healthy Italian population aged 25–34 years, at the time of the study, the rate of employment was 63.0% and that of unemployment 14.7%. In particular, in Piedmont (the region of northern Italy where our center is based), the occupational rate for young adults (25–34 years) in 2019 was 70.3% [21]. We did not find significant differences in occupational rate according to gender (males 59.8%, females 57.1%). Occupation rate of our CCS is similar to that of the general Piedmont population in females (57.8%), but lower in males (71.4%) [21]. Furthermore, the slightly better occupational rate of our CCS compared to young adults of the general population in Piedmont (75.41% vs. 70.3%) was similar to data reported in a case study in Germany [22].

Occupation rates significantly vary by the considered country. In North Europe (Denmark, Finland, Sweden) at 30 years of age, 6.7% of CCS are unemployed (the population comparison unemployed is 6.6%) [1]. In France, the employment rate of CCS is around 79%, and the health-related unemployment rate was significantly different from the general population only for survivors of brain tumor (4% vs. 28%) and not for other diagnoses [13]. In Great Britain, the percentage of CCS employed is 57.7% for females and 67.2% for males [12]. These observations highlight the importance of considering the country-specific vocational system.

According to groups of the ISCO-08 [16], the prevalence of most categories (i.e., managers, professionals, services and sales worker, plant and machine operator) in our CCS was comparable to that observed by Frederiksen et al. in their population [1]. However, other categories (i.e., clerical and support worker, skilled agricultural, forestry and fishery workers or armed forces occupation) had no or little representation in our population but are more frequent in the North European cohort. This discrepancy can be likely explained by the small number of participants of our study, but also by differences in the socio-economic scenario among Italy and North European countries.

Several studies already highlighted that the main risk factors for unemployability in CCS concern health area, and CNS tumor survivors and subjects diagnosed before 15 years of age are reported as the most compromised categories [1,10,11]. These observations likely reflect the impact of physical late effects on the ability to find and keep a job. Our results confirm the relationship between the prevalence of late effects and unemployment. The strength of this association increases with increasing severity of late effects, pointing out the impact of physical late effects on psychosocial functioning of survivors. We did not observe a significant association between sex and age and employment.

Besides the objective evaluation of the occupational status, it is also interesting to consider the subjective perception of CCS on their job and financial situation. Our results showed that being an unemployed survivor with comorbidities significantly worsens the satisfaction on occupation and economic situation, when compared to workless without late effects. This negative perception was also revealed when they were compared to occupied survivors (employed or students), with or without late effects. Similarly, Soejima and colleagues underlined that the physical late effects have a negative impact on the subjective perception of worries about employment among unemployed survivors [5].

This study has some limitations. First, the relatively small sample size could negatively impact estimate precision. The participation rate to the study (82%; 51 out of 291 did not accept participation) can reduce the representativeness of our results. Moreover, our observations reflect only the northern Italy socio-economic context. Nevertheless, our results give a comprehensive description of the occupational status of our sample of CCS, suggesting socio-economic vulnerable subgroups of survivors that should be strictly monitored. Evidence and recommendations suggest that in a routine follow-up, a healthcare provider should be dedicated to the surveillance of psychosocial development [8]. Young adult survivors might benefit from tailored interventions (support, vocational orientation, detection of potential health problems that can interfere in obtaining and maintaining a job) to help them obtain jobs suitable for their health.

## 5. Conclusions

Our results underlined risk factors for occupation status of young adult CCS. Vulnerable categories of CCS (brain tumor survivors and those with severe late effects) should be surveilled in medical as well as in socioeconomic aspects.

Future research in this field can be focused on interventions to improve employment status. It would be also interesting to implement a longitudinal monitoring of occupational status according to actual social conditions (i.e., how the COVID-19 pandemic impacted the working area of cancer survivors). To complete the description of occupational status of CCS in the Italian scenery, a multicenter study could be implemented.

## Figures and Tables

**Figure 1 cancers-14-03984-f001:**
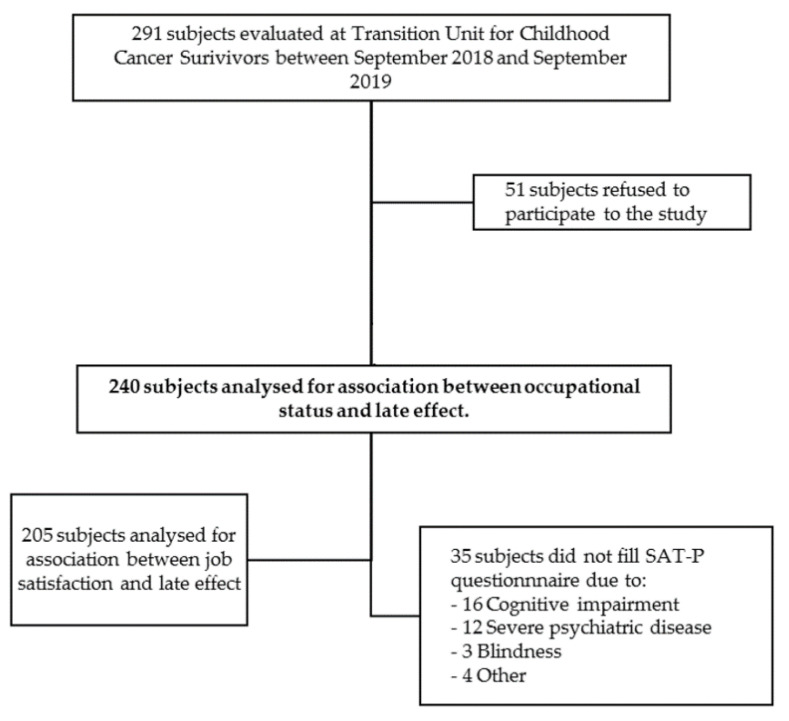
Inclusion process of participants.

**Figure 2 cancers-14-03984-f002:**
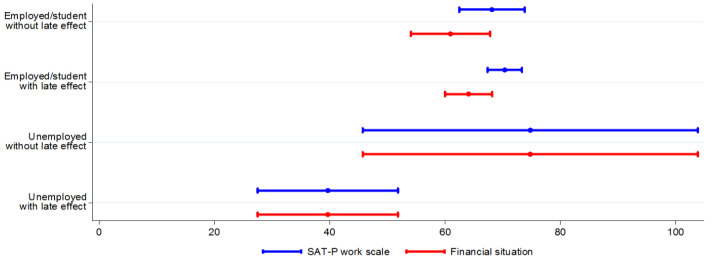
Work and financial situation satisfaction, according to occupational status and to the presence of late effect (mean).

**Table 1 cancers-14-03984-t001:** Socio-demographic and clinical characteristics according to occupational status.

	Occupational Status						Total	
	Student		Unemployed		Employed			
	No.	%	No.	%	No.	%	No.	%
**Sex**								
Female	25	47.17	17	36.96	56	39.72	98	40.83
Male	28	52.83	29	63.04	85	60.28	142	59.17
**Age at the time of the study (years)**								
18–24	40	75.47	17	36.96	37	26.24	94	39.17
25–29	13	24.53	14	30.43	50	35.46	77	32.08
≥30	0	0	15	32.61	54	38.3	69	28.75
**Marital Status**								
Single	39	73.58	36	78.26	52	36.88	127	52.92
Partnership	14	26.42	10	21.74	68	48.23	92	38.33
Married	0	0	0	0	19	13.47	19	7.92
Separated	0	0	0	0	2	1.42	2	0.83
**Offspring**								
No	53	100	43	93.48	128	90.78	224	93.34
Yes	0	0	3	6.52	13	9.22	16	6.66
**Age at the first cancer diagnosis (years)**								
0–4	14	26.42	15	32.61	35	24.82	64	26.67
5–9	15	28.3	13	28.26	33	23.4	61	25.42
10–18	24	45.28	18	39.13	73	51.77	115	47.92
**Period (of the first cancer diagnosis)**								
1985–1989	0	0	2	4.35	8	5.67	10	4.17
1990–1999	5	9.43	17	36.96	51	36.17	73	30.42
2000–2012	48	90.57	27	58.7	82	58.16	157	65.42
**Cancer diagnosis**								
*Hematologic Malignancies*	41	77.36	33	71.74	100	70.92	174	72.50
Acute Lymphoblastic Leukemia	19	35.85	15	32.61	53	37.59	87	36.25
Hodgkin Lymphoma	5	9.43	6	13.04	26	18.44	37	15.42
Non-Hodgkin’s Lymphoma	5	9.43	3	6.52	13	9.22	21	8.75
Acute Myeloid Leukemia and myelodysplastic syndrome	11	20.75	8	17.39	7	4.96	26	10.83
Other hematological not specified	1	1.89	1	2.17	1	0.71	3	1.25
*Brain tumors*	6	11.32	9	19.57	14	9.93	29	12.08
*Sarcomas*	6	11.32	2	4.35	18	12.77	26	10.83
*Others*	0	0	2	4.35	9	6.38	11	4.58
**Any radiation**								
No	31	58.49	14	30.43	71	50.35	116	48.33
Yes	22	41.51	32	69.57	70	49.65	124	51.67
**Cranial irradiation**								
No	43	81.13	29	63.04	117	82.98	189	78.75
Yes	10	18.87	17	36.96	24	17.02	51	21.25
**Any chemotherapy**								
No	2	3.77	1	2.17	2	1.42	5	2.08
Yes	51	96.23	45	97.83	139	98.58	235	97.92
**Hematopoietic Stem Cell Transplantation**								
No	36	67.92	26	56.52	105	74.47	167	69.58
Yes	17	32.08	20	43.48	36	25.53	73	30.42
**Endocrinological late effects**								
No	24	45.28	8	17.39	63	44.68	95	39.58
Moderate	28	52.83	28	60.87	73	51.77	129	53.75
Severe	1	1.89	10	21.74	5	3.55	16	5.5
**Cardiovascular late effects**								
No	43	81.13	26	56.52	104	73.76	173	72.08
Moderate	10	18.87	20	43.48	36	25.53	66	27.50
Severe	0	0	0	0	1	0.71	1	0.42
**Pulmonary late effects**								
No	52	98.11	45	97.83	139	98.58	236	98.33
Moderate	1	1.89	0	0	1	0.71	2	0.83
Severe	0	0	1	2.17	1	0.71	2	0.83
**Neurological late effects**								
No	50	94.34	30	65.22	124	87.94	204	85.00
Moderate	2	3.77	7	15.22	12	8.51	21	8.75
Severe	1	1.89	9	19.57	5	3.55	15	6.25
**Musculoskeletal late effects**								
No	47	88.68	42	91.3	126	89.36	215	89.58
Moderate	3	5.66	4	8.7	13	9.22	20	8.33
Severe	3	5.66	0	0	2	1.42	5	2.08
**Reproductive/genital late effects**								
No	31	58.49	22	47.83	95	67.38	148	61.67
Moderate	5	9.43	10	21.74	6	4.26	21	8.75
Severe	17	32.08	14	30.43	40	28.37	71	29.58
**Other late effects ***								
No	48	90.57	31	67.39	121	85.82	200	83.33
Moderate	5	9.43	11	23.91	12	8.51	28	11.67
Severe	0	0	4	8.7	8	5.67	12	5.00
**Any Late Effect**								
No	13	24.53	5	10.87	34	24.11	52	21.67
Yes	40	75.47	41	89.13	107	75.89	188	78.33
**Late effects intensity**								
No late-effects	13	24.53	5	10.87	34	24.11	52	21.67
At least one moderate and no severe	21	39.62	16	34.78	53	37.59	90	37.50
At least one severe	19	35.85	25	54.35	54	38.3	98	40.83
**Total**	53	100	46	100	141	100	240	100.00

* Auditory-hearing, gastrointestinal, hepatobiliary, hematologic, immunologic, infectious, ocular/visual, renal/urinary late effects.

**Table 2 cancers-14-03984-t002:** Occupational positions, according to the International Standard Classification of Occupations version 08 (ISCO-08).

Occupation	No.	%
(1) Managers	5	3.55
(2) Professionals	30	21.28
(3) Technicians and associate professionals	35	24.82
(4) Clerical support workers	1	0.71
(5) Services and sales workers	39	27.66
(6) Skilled agricultural, forestry and fishery workers	0	0.00
(7) Craft and related trades workers	17	12.06
(8) Plant and machine operators, and assemblers	10	7.09
(9) Elementary occupation	4	2.84
(10) Armed forces occupation	0	0.00
**Total**	141	100.00

**Table 3 cancers-14-03984-t003:** Crude and adjusted effects on unemployment.

	All Patients (N = 240)						Only Employed and Unemployed Participants (N = 187)					
	Univariate Model			Multivariate Model			Univariate Model			Multivariate Model		
	OR	95% CI	*p* Value	OR	95% CI	*p* Value	OR	95% CI	*p* Value	OR	95% CI	*p* Value
**Sex**												
**Male**	1	[1.00,1.00]	.	1	[1.00,1.00]	.	1	[1.00,1.00]	.	1	[1.00,1.00]	.
**Female**	0.82	[0.42,1.59]	0.552	0.83	[0.42,1.62]	0.579	0.89	[0.45,1.77]	0.739	0.90	[0.44,1.82]	0.766
**Age at the time of the study (continuous)**	1.02	[0.95,1.09]	0.560	1	[0.93,1.08]	0.967	0.96	[0.89,1.03]	0.249	0.94	[0.87,1.01]	0.103
**Age at the** **time of the study** **(years)**												
**18–24**	1	[1.00,1.00]	.				1	[1.00,1.00]	.			
**25–29**	1.01	[0.46,2.20]	0.987				0.61	[0.27,1.39]	0.239			
**≥30**	1.26	[0.58,2.74]	0.562				0.6	[0.27,1.36]	0.224			
**Any Late Effects**												
**No**	1	[1.00,1.00]	.	1	[1.00,1.00]	.	1	[1.00,1.00]	.	1	[1.00,1.00]	.
**Yes**	2.62	[0.98,7.02]	0.055	2.61	[0.96,7.08]	0.060	2.61	[0.95,7.12]	0.062	2.96	[1.06,8.26]	**0.038**

Abbreviations: CI: confidence interval, OR: odds ratio.

**Table 4 cancers-14-03984-t004:** Crude and adjusted effect on Work subscale of SAT-P.

	All Patients (N = 205) *						Only Employed and Unemployed Participants (N = 156)					
	Univariate Model			Multivariate Model			Univariate Model		Multivariate Model			
	Linear Coeff	95% CI	*p* Value	Linear Coeff	95% CI	*p* Value	Linear Coeff	95% CI	*p* Value	Linear Coeff	95% CI	*p* Value
**Sex Female**	−1	[−8.01,6.01]	0.780	−0.65	[−7.40,6.09]	0.849	2.04	[−6.41,10.48]	0.635	2.76	[−5.32,10.84]	0.501
**Age at the time of the study (continuous)**	0	[−0.76,0.75]	0.992	0.11	[−0.63,0.85]	0.778	0.09	[−0.83,1.01]	0.847	0.11	[−0.80,1.02]	0.809
**Age at the time of the study (years)**												
**25–29**	0.01	[−8.16,8.18]	0.997				1.28	[−9.14,11.71]	0.808			
**≥30**	−2.63	[−11.23,5.97]	0.547				−0.93	[−11.26,9.41]	0.86			
**Any late effects**	−1.72	[−9.83,6.39]	0.676				−3.67	[−13.32,5.98]	0.454			
**Unemployed**	−17.15	[−27.10,−7.20]	0.001	13.94	[−8.53,36.40]	0.223	−16.95	[−27.58,−6.32]	0.002	13.84	[−9.98,37.66]	0.253
**Any late effect** **(employed or students)**				2.94	[−5.47,11.35]	0.492				2.36	[−7.84,12.56]	0.648
**Any late effect (unemployed)**				−35.47	[−59.19,−11.74]	0.004				−35.67	[−60.41,−10.94]	0.005

Abbreviations: CI: confidence interval, Coeff: coefficient. * Only participants who filled SAT-P questionnaire.

**Table 5 cancers-14-03984-t005:** Crude and adjusted effect on SAT-P financial satisfaction item.

	All Patients (N = 205) *						Only Employed and Unemployed Participants (N = 156)					
	Univariate Model			Multivariate Model			Univariate Model			Multivariate Model		
	Linear Coeff	95% CI	*p* Value	Linear Coeff	95% CI	*p* Value	Linear Coeff	95% CI	*p* Value	Linear Coeff	95% CI	*p* Value
**Female Sex**	−4.5	[−10.58,1.58]	0.146	−3.99	[−9.55,1.57]	0.159	−2.20	[−9.32,4.93]	0.544	−1.28	[−7.54,4.98]	0.687
**Age at the time of the study (continuous)**	0.15	[−0.51,0.80]	0.654	0.29	[−0.32,0.90]	0.357	−0.06	[−0.83,0.72]	0.888	−0.08	[−0.78,0.62]	0.821
**Age at the** **time of the study** **(years)**												
**25–29**	−1.29	[−8.42,5.84]	0.721				−2.93	[−11.71,5.86]	0.511			
**≥30**	−0.28	[−7.78,7.23]	0.942				−2.31	[−11.03,6.40]	0.601			
**Late effects intensity**												
**At least one moderate and** **no**	−2.37	[−10.19,5.45]	0.550				−4.53	[−13.51,4.46]	0.321			
**At least one severe**	−3.79	[−11.72,4.15]	0.348				−6.01	[−15.18,3.15]	0.197			
**Any late effects**	−3.05	[−10.11,4.01]	0.395				−5.23	[−13.35,2.88]	0.204			
**Unemployed**	−22.91	[−31.25,−14.58]	<0.001	7.46	[−11.06,25.99]	0.428	−24.90	[−33.26,−16.54]	<0.001	4.52	[−13.93,22.97]	0.629
**Any late effects** **(employed or students)**				1.45	[−5.49,8.39]	0.681				1.49	[−6.42,9.39]	0.711
**Any late effects** **(unemployed)**				−35.94	[−55.51,−16.37]	<0.001				−34.76	[−53.92–15.59]	<0.001

Abbreviations: CI: confidence interval, Coeff: coefficient. * Only participants who filled SAT-P questionnaire.

## Data Availability

The data presented in this study are available on request from the corresponding author.

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
