# Peer review of "Work Placement and Job Satisfaction in Long-Term Childhood Cancer Survivors: The Impact of Late Effects"

_cancers, 2022, doi:10.3390/cancers14163984_

Round 1

Reviewer 1 Report

this paper is interesting and can be improved by adding some comments/points : 

- is it correct to administer a " Job satisfaction questionnaire" to  unemployed survivors? please specify .

- "The presence of at least one severe late effect was significantly associated with the probability of unemployment ": please specify the type of tumor and/or type of  treatment most related with severe late effect/s.

- do you have any information on family' s income and/or cultural background? these data could  be related with academic degree and employement.

- please add a sentence about gender and job placement ;

- do you have any information on marital status and pregnacies ? how many survivors have children ? please specify how many survivors are housewives.

- are there any difference between survivors living in town compared to those living in a rural area?

-line 175-176. this sentence require an explanation or a comment .

Author Response

This paper is interesting and can be improved by adding some comments/points: 

- is it correct to administer a " Job satisfaction questionnaire" to unemployed survivors? please specify.

We thank the Reviewer for this comment. We confirm that it is correct to administer the SAT-P questionnaire also to unemployed survivors. Indeed, the questionnaire is validated to evaluate no exclusively the job-related satisfaction but the satisfaction about the quality of daily life. In the case of unemployed subjects, as detailed in the methods section, we considered only the answer to the item about financial situation, that could be independent from the occupational status.

- "The presence of at least one severe late effect was significantly associated with the probability of unemployment ": please specify the type of tumor and/or type of  treatment most related with severe late effect/s.

We thank the Reviewer for this comment. We considered the impact of late effects on unemployment as good proxy of combination between cancer diagnosis and cancer treatments effect. As expected, we observed an association between cancer diagnosis and late effect: in patients with a pediatric brain tumor or sarcoma the proportion of at one severe late effect was higher than in other patients. For the Reviewer convenience, we summarized this results in the table below.

Late effect

None

At least one moderate and none severe

At least one severe

Total

N

%Row

N

%Row

N

%Row

N

Hematologic

43

24.71

70

40.23

61

35.06

174

Brain tumor

1

3.45

8

27.59

20

68.97

29

Sarcoma

4

15.38

8

30.77

14

53.85

26

Other

4

36.36

4

36.36

3

27.27

11

Total

52

21.67

90

37.50

98

40.83

240

- do you have any information on family' s income and/or cultural background? these data could be related with academic degree and employement.

We thank the Reviewer for this interesting suggestion but, unfortunately, we did not have information about family income and or cultural background of participants.

- please add a sentence about gender and job placement;

We added a comment in the Discussion section of the manuscript, as suggested.

- do you have any information on marital status and pregnacies? how many survivors have children ? please specify how many survivors are housewives.

Thank you for this interesting suggestion. We added in table 1 data about marital status and offspring.

None of the included patients was housewife at the time of the study. Unemployed survivors were all looking for a job or unable to do a job. So, they cannot be considered housewives in any case

- are there any difference between survivors living in town compared to those living in a rural area?

We thank the Reviewer for this interesting suggestion but, unfortunately, we did not have information about survivor’s area of residence.

-line 175-176. this sentence require an explanation or a comment.

We thank the reviewer for this comment, and the text has been amended to clarify this point. The sentence has been removed, and the issue have been discussed in lines 235-239.

Reviewer 2 Report

We thank the authors for this interesting manuscript on work placement and job satisfaction in long-term CCS and the impact of late effects.

Line 29 240 CCS is a small number of participants, please consider to indicate and discuss the participation rates of 82% and 70% (291/ -51/ -35) 

Line 32 p <0.05, Line 34 p = 0.004, .. please consider to give p-values uniformly, for example, the same number of decimal places

Line 32,33 please explain b, CI, OR  e.g. under a table as a footnote

Line 35 Clarify whether occupation or re-occupation is meant?

Line 43 pediatric, Line 9 paediatric .. please use consistent British or American English

Line 66 please specify the risk factor “younger age at diagnosis”, please indicate the groups for age at diagnosis in materials and methods

Line 65-69 sentence is very long, please consider to make two sentences

Line 67 please consider to say “such as” instead of “as well as”, please consider having a native speaker read through the manuscript

Line 77 please consider to add some more information on the Transition Unit e.g. since when does it exist

Line 116 please specify how the interaction between the two last variables (late effect and occupational status) was analysed

Line 126  “most participants” with 39.17% in age group 18-24 does not sound quite correct

Line 126,127 please consider to add groups for age at study (and age at diagnosis) in chapter “materials and methods”

127 use “were” instead of “was” aged

Table one:

·       The groups “age at time of the study” and “age at first cancer diagnosis” include different numbers of years, please consider to comment on that

·       Please consider to remove the line below the Hematologic Malignancies

Table 3,4,5 please improve format, explain the difference between “crude” and “adjusted” effect

Line 165 please consider to change: when they were affected

Line 176 There is a dot missing at the end of the sentence.

Line 179 vs 183 In the present study, the occupation rate in CCS compared to young adults in the  general population in Piedmont was 75.41% vs. 70.3 %. Consider to underline that the occupation rate in CCS was slightly better; this was also the case in a study in Germany (Dieluweit U, Debatin KM, Grabow D et al. Educational and vocational achievement among long-term survivors of adolescent cancer in Germany. Pediatr Blood Cancer. 2011; 56(3):432-8.)

Results: Compared to peers from the general population, survivors of cancer during adolescence achieved higher educational and vocational levels. A higher proportion of survivors was employed; however, survivors were significantly older when starting their first occupation. Subgroup analyses revealed that neuropsychological late effects were associated with reduced rates of graduation from university and of employment among the survivors. No such effect of neuro-cognitive late effects occurred for high school graduation.

Please discuss, maybe already in the introduction, a possible “post traumatic growth”

Line 186-188 Please also give the occupation rates in the general population in France and in GB and if possible consistently: compare either occupation rate or employment rate or unemployment rate for better comparability.

Line 209 Please consider to change: when they were compared

Line 210 Please consider to change: Soejima and Colleagues

Line 228 Please check grammar

Author Response

We thank the authors for this interesting manuscript on work placement and job satisfaction in long-term CCS and the impact of late effects.

Line 29 240 CCS is a small number of participants, please consider to indicate and discuss the participation rates of 82% and 70% (291/ -51/ -35)      

We thank the Reviewer for the comment. A comment has been included within Discussion (lines 244-245).

Line 32 p <0.05, Line 34 p = 0.004, .. please consider to give p-values uniformly, for example, the same number of decimal places

In the revised version of the manuscript p-values are given uniformly.

Line 32,33 please explain b, CI, OR e.g. under a table as a footnote

In the revised version we added footnote to clarify.

Line 35 Clarify whether occupation or re-occupation is meant?

Participants were diagnosed with cancer in childhood or adolescence, so we only consider occupation and not re-occupation in our research.

Line 43 pediatric, Line 9 paediatric .. please use consistent British or American English

Thank you for the suggestion, we checked and modified the text.

Line 66 please specify the risk factor “younger age at diagnosis”, please indicate the groups for age at

diagnosis in materials and methods

Thank you for your suggestion. In the Introduction section (line 66), we were reporting findings in literature related to “younger age at diagnosis” as a risk factor for unemployment (Frederiksen et al., 2021; Brinkman et al., 2018; Kirchoff et al., 2010), so it is impossible to specify the ages of groups considered in different studies. As far as our results, we better define the age groups in the methods.

Line 65-69 sentence is very long, please consider to make two sentences

Thank you for the suggestion. The text has been amended accordingly.

Line 67 please consider to say “such as” instead of “as well as”, please consider having a native speaker

read through the manuscript

Thank you for the suggestion. The text has been amended accordingly.

Line 77 please consider to add some more information on the Transition Unit e.g. since when does it exist

Thank you for the suggestion. In the new version of the manuscript we provided a brief description of our Unit (lines 83-93).

Line 116 please specify how the interaction between the two last variables (late effect and occupational status) was analysed

Generally, in a statistical model, a term of interaction was included when there is the hypothesis that an independent variable has a different effect on the outcome depending on the values of another independent variable.

In this study, we hypothesized that late effects were differently associated to the work scale of SAT-P. Model estimates confirmed this hypothesis: we observed a strong association between late effect and work scale of SAT-P in unemployed CCS and none association between late effect and satisfaction in employed CCS.

Line 126  “most participants” with 39.17% in age group 18-24 does not sound quite correct

Thank you for the suggestion. The text has been amended accordingly.

Line 126,127 please consider to add groups for age at study (and age at diagnosis) in chapter “materials and methods”

We modified Material and Method section according with this observation.

127 use “were” instead of “was” aged

Thank you for the suggestion. The text has been amended accordingly.

Table one:

     The groups “age at time of the study” and “age at first cancer diagnosis” include different numbers of         years, please consider to comment on that.

Thank you for this observation. We divided age at the time of the study and age at diagnosis in different groups on the basis of a clinical consideration, particularly on cognitive and psychological development of participants. We tried to summarize different developmental stages that imply different ways of understanding, experience and coping with the event.

       Please consider to remove the line below the Hematologic Malignancies

Thank you for the suggestion. The table has been amended accordingly.

Table 3,4,5 please improve format, explain the difference between “crude” and “adjusted” effect-

In order to improve the usability of tables we substituted the terms crude and adjusted effect with their synonymous “univariate” and “multivariate” model.

Line 165 please consider to change: when they were affected

Thank you for the suggestion. The text has been amended accordingly.

Line 176 There is a dot missing at the end of the sentence.

Thank you for the suggestion. The text has been amended accordingly.

Line 179 vs 183 In the present study, the occupation rate in CCS compared to young adults in the general population in Piedmont was 75.41% vs. 70.3 %. Consider to underline that the occupation rate in CCS was slightly better; this was also the case in a study in Germany (Dieluweit U, Debatin KM, Grabow D et al. Educational and vocational achievement among long-term survivors of adolescent cancer in Germany. Pediatr Blood Cancer. 2011; 56(3):432-8.)

Results: Compared to peers from the general population, survivors of cancer during adolescence achieved higher educational and vocational levels. A higher proportion of survivors was employed; however, survivors were significantly older when starting their first occupation. Subgroup analyses revealed that neuropsychological late effects were associated with reduced rates of graduation from university and of employment among the survivors. No such effect of neuro-cognitive late effects occurred for high school graduation.

Thank you for your interesting suggestion. We implemented it in the discussion section (lines 207-209) and added the reference [22].  

Please discuss, maybe already in the introduction, a possible “post traumatic growth”

Thank you for your suggestion. We added a section in the Introduction about the possible positive attitude that young adult survivors can experience into key developmental tasks (lines 73-75).

Line 186-188 Please also give the occupation rates in the general population in France and in GB and if possible consistently: compare either occupation rate or employment rate or unemployment rate for better comparability.

Thank you for your suggestion. We added this information for France population in the Discussion section (lines 213-215). Unfortunately, it was impossible to find the original data used by Frobisher for the comparison between its cohort of CCS and the British general population.

Line 209 Please consider to change: when they were compared

Thank you for the suggestion. The text has been amended accordingly.

Line 210 Please consider to change: Soejima and Colleagues

Thank you for the suggestion. The text has been amended accordingly.

Line 228 Please check grammar

Thank you for the suggestion. Grammar has been checked.